# Combining Low-Dose Computer-Tomography-Based Radiomics and Serum Metabolomics for Diagnosis of Malignant Nodules in Participants of Lung Cancer Screening Studies

**DOI:** 10.3390/biom14010044

**Published:** 2023-12-28

**Authors:** Joanna Zyla, Michal Marczyk, Wojciech Prazuch, Magdalena Sitkiewicz, Agata Durawa, Malgorzata Jelitto, Katarzyna Dziadziuszko, Karol Jelonek, Agata Kurczyk, Edyta Szurowska, Witold Rzyman, Piotr Widłak, Joanna Polanska

**Affiliations:** 1Department of Data Science and Engineering, Silesian University of Technology, 44-100 Gliwice, Poland; joanna.zyla@polsl.pl (J.Z.); wojciechprazuch3@gmail.com (W.P.); joanna.polanska@polsl.pl (J.P.); 2Yale Cancer Center, Yale School of Medicine, New Haven, CT 06510, USA; 3Department of Thoracic Surgery, Medical University of Gdansk, 80-210 Gdansk, Poland; magdalena.sitkiewicz@gumed.edu.pl (M.S.); agata.durawa@gumed.edu.pl (A.D.); wrzyman@gumed.edu.pl (W.R.); 42nd Department of Radiology, Medical University of Gdansk, 80-210 Gdansk, Poland; jelitto@gumed.edu.pl (M.J.); katarzyna.dziadziuszko@gumed.edu.pl (K.D.); eszurowska@gumed.edu.pl (E.S.); piotr.widlak@gumed.edu.pl (P.W.); 5Center for Translational Research and Molecular Biology of Cancer, Maria Sklodowska-Curie National Research Institute of Oncology, Gliwice Branch, 44-100 Gliwice, Poland; karol.jelonek@gliwice.nio.gov.pl; 6Department of Biostatistics and Bioinformatics, Maria Sklodowska-Curie National Research Institute of Oncology, Gliwice Branch, 44-100 Gliwice, Poland; agata.kurczyk@gliwice.nio.gov.pl

**Keywords:** classification models, integration, early detection, lung cancer, screening study

## Abstract

Radiomics is an emerging approach to support the diagnosis of pulmonary nodules detected via low-dose computed tomography lung cancer screening. Serum metabolome is a promising source of auxiliary biomarkers that could help enhance the precision of lung cancer diagnosis in CT-based screening. Thus, we aimed to verify whether the combination of these two techniques, which provides local/morphological and systemic/molecular features of disease at the same time, increases the performance of lung cancer classification models. The collected cohort consists of 1086 patients with radiomic and 246 patients with serum metabolomic evaluations. Different machine learning techniques, i.e., random forest and logistic regression were applied for each omics. Next, model predictions were combined with various integration methods to create a final model. The best single omics models were characterized by an AUC of 83% in radiomics and 60% in serum metabolomics. The model integration only slightly increased the performance of the combined model (AUC equal to 85%), which was not statistically significant. We concluded that radiomics itself has a good ability to discriminate lung cancer from benign lesions. However, additional research is needed to test whether its combination with other molecular assessments would further improve the diagnosis of screening-detected lung nodules.

## 1. Introduction

Lung cancer holds the unenviable position of being the leading cause of cancer-related fatalities in both men and women. It accounts for 21% of all cancer deaths among men, followed by prostate (11%) and colon and rectum cancers (9%). In women, it represents 21% of all cancer deaths, followed by breast (15%) and colon and rectum cancers (8%) [1]. In the year 2020, the global landscape witnessed a staggering 2.2 million new cases of lung cancer, resulting in 1.8 million fatalities attributed to this serious disease [2]. Notably, in Poland, 63 individuals succumb to lung cancer each day, ranking high incidence in Europe. Approximately, 81% of lung cancer deaths result from direct cigarette smoking, with indirect smoking and air pollution being the following contributing factors. In Poland, the population of smokers is estimated at a substantial 8 million individuals. The low survival rate for lung cancer is mainly related to late diagnosis. An asymptomatic course of disease progression causes a low rate of early-stage detection. In the case of symptoms, long-term smoking patients usually develop chronic symptoms similar to early signs of lung cancer earlier but often disregard them if they are not severe or present at all. Moreover, patients’ beliefs and worries about health changes that may indicate lung cancer appear to play a crucial part in the delay in diagnosis. Late diagnosis and oncological therapy often fail to cure the patient. Still, they could do so in the earlier stages of the disease, for which the survival rate is much higher [3].

The diagnosis of early lung cancer in the real world typically initiates with X-ray chest radiography or computed tomography (CT) performed for other reasons. Nowadays, the systemic approach to the detection of early-stage lung cancer has been implemented in a few countries based on the results of two pivotal randomized clinical trials: NLST and NELSON. These clinical trials provided incontrovertible evidence of reducing lung-cancer-related mortality by 20% and 26%, respectively, with the implementation of screening programs involving low-dose computed tomography (LDCT), specifically targeting high-risk groups, such as tobacco smokers [4,5]. LDCT-based lung cancer screening leads to thousands of images with a substantial number of lung nodules that should be appropriately categorized. Hence, radiologists play a crucial role in identifying nodules or tumors within the lung parenchyma and predicting the risk of malignancy. If any nodule displays malignant features, the process of invasive lung cancer diagnosis begins. 

Lung cancer screening has been established to enhance the rate of early lung cancer detection in cases where the disease remains asymptomatic during its progression. These attempts target a high-risk demographic comprising middle-aged and elderly individuals with a history of long-term smoking. Ideally, lung cancer screening rounds should occur annually, which places a significant burden on current radiological resources. The evaluation of image-based screening for detecting subtle abnormalities can be a complex and time-consuming process. Moreover, a significant proportion of false positive results of CT-based tests affect lung cancer diagnosis. It is assumed that the diagnostic accuracy of cancer detection could be increased by supplementing low-dose CT imaging with additional diagnostic tests, particularly molecular biomarkers [6,7]. Among the hypothetical biomarkers of early lung cancer that could complement CT-based diagnosis are different molecular and cellular components of blood [8,9,10,11]. Metabolites present in blood are promising candidates for biomarkers since they could be potentially detected through “liquid biopsy” [12]. For example, choline-containing phospholipids and sphingolipids are serum/plasma components that discriminate between lung cancer patients and healthy individuals [13,14,15]. Recently, we performed a metabolomics study to search for serum metabolites that differentiated three groups of lung cancer screening participants: patients with screen-detected lung cancer, individuals with benign pulmonary nodules, and those without any lung alterations. However, despite several specific compounds having significant differences among compared groups, the low accuracy of classification models was observed (AUC = 60%) due to substantial heterogeneity in the levels of analyzed metabolites [16]. 

Radiomics allows the extraction of a comprehensive set of features from an LDCT image for automated cancer detection and the diagnosis of malignant lesions [17]. We hypothesized that the combination of disease-related features observed at the systemic level (i.e., the features of serum metabolome) and a depiction of local pathological changes (i.e., the features of LDCT images) would enhance the precision of lung cancer classification models. To prove our concept, we developed a method for differentiating between benign and malignant nodules detected in lung cancer screening participants by combining the results of LDCT and metabolomic modalities. We gathered data from two different screening cohorts and tested two machine learning models and several integration algorithms to find the best solution.

## 2. Materials and Methods

### 2.1. Study Subject

Material included in this study was collected during two lung cancer screening programs performed by the Medical University of Gdansk in the years 2009–2011 (PPPBWWRP—Pomorski Pilotażowy Program Badań Wczesnego Wykrywania Raka Płuca) and 2016–2018 (MOLTEST-BIS) [3,16]. These programs enrolled more than 14 thousand participants and offered LDCT examinations for current or former smokers with at least a 20-pack-year history, aged from 50 to 75 years. This report involves two groups of participants of the project: (i) individuals with CT-detected lung nodules that were confirmed benign via histopathology, further marked as benign, and (ii) patients who were ultimately diagnosed with lung cancer, further marked as malignant. For those patients, two types of measurements (modalities) were collected: (i) radiomic characteristics from LDCT scans (1086 participants of either PPPBWWRP or MOLTEST-BIS dataset), and (ii) serum metabolome profiles (246 participants of the MOLTEST-BIS cohort). Several regions of suspicious elements from the LDCT scan were collected for one of the patients. The basic characteristics of the analyzed groups are presented in Table 1. Studies were approved by the appropriate Ethics Committees (the Medical University of Gdansk, approval nos. NKEBN/42/2009 and NKBBN/376/2014), and all participants provided informed consent indicating their voluntary participation in the project and provision of blood samples for future research.

### 2.2. Metabolomic Data

The detailed procedure of data collection, preparation, and preprocessing is described elsewhere [16]. Briefly, for serum samples isolated from the peripheral blood of MOLTEST-BIS participants, the measures were obtained via a high-resolution mass spectrometry assay using an Absolute IDQ p400 HR kit (test plates in the 96-well format; Biocrates Life Sciences AG, Innsbruck, Austria) according to the manufacturer’s protocol. For obtained measurements, batch correction and missing data imputation were performed as described elsewhere [16]. Finally, concentrations of 259 specific metabolites (or lipid isomer groups) and aggregated concentrations of different metabolite classes: acylcarnitines, amino acids, biogenic amines, glycerophospholipids, sphingolipids, cholesterol esters (CEs), glycerides, triglycerides (TGs), diglycerides (DGs), phosphatidylcholines (PCs), lysophosphatidylcholines (LPCs), and total lipids were analyzed; this resulted in 271 metabolomics features (Appendix A).

### 2.3. Radiomic Data

Radiomic data for this project come from measurements performed on 1086 patients (925 from PPPBWWRP and 161 from MOLTEST-BIS). For each patient, the LDCT was performed, and for abnormalities found in the lung parenchyma, an annotation was prepared by an expert radiologist. Regions with abnormalities were categorized into the following groups: cancer, suspicious nodules, inflammation, benign nodules, lymph nodes, fibrosis, and calcification (one patient could have different pathologies classified in different groups). Cancer, suspicious nodules, and inflammation were considered “malignant”, while the remaining categories represent the “benign” group. Next, the mask of objects and their segmentations were extracted using a multi-step pre-processing and segmentation algorithm [18]. For extracted regions of the lung, the radiomics features were calculated with the usage of the PyRadiomics package version 3.0 [19]. In total, 107 radiomics features were analyzed for 5180 fragments of annotated images from analyzed patients (Appendix A). Finally, radiomics features were internally standardized (scaled and shifted) using non-parametric statistics, like median and interquartile range, calculated on benign samples within a cohort.

### 2.4. Univariate Analysis

Each analyzed feature from both modalities was tested due to the normality of distribution using the Shapiro–Wilk test. As the data were highly skewed, to estimate the significance of differences in analysis groups, the Mann–Whitney test was used. Moreover, each feature’s biserial correlation (r_g_) was calculated and treated as an effect size measure. Finally, the Benjamini–Hochberg procedure for the FDR correction was applied when necessary [20]. All statistical hypotheses were tested at the 5% significance level.

### 2.5. Machine Learning Sets

The classification models were constructed using two different machine learning (ML) approaches: (i) logistic regression (LR) and (ii) random forest (RF). Both ML methods were performed on the same training and test sets. The test set was extracted from the MOLTEST-BIS cohort by taking 20 benign and 20 malignant cases for which both radiomic and metabolomic data were available. Moreover, for radiomic data, the test set was expanded for additional cases from the PPPWWRP cohort. The remaining samples were gathered into a training set. A summary of the training and test set for both omics is presented in Table 2.

### 2.6. Logistic Regression Models 

For the initially selected train set multiple random cross-validation (MRCV) was performed for logistic regression (LR), as follows: (i) data were split into training and validation (70%/30%) subsets; (ii) forward feature selection was performed on training subset with ΔBIC ≤ 2 as a stop criterion; (iii) an evaluation of classification parameters was conducted on the training and validation subsets; and (iv) an estimation of classification threshold was made by maximizing balanced accuracy (BAcc). The MRCV procedure was repeated 100 times. Next, feature ranking was generated as follows: (i) the features included in each model were sorted by their order of addition in the forward procedure; (ii) the proportional order was multiplied by the BAcc of the validation set at a particular fold; and (iii) the elbow technique was used to extract the most relevant features for the final LR model. Finally, the model was built on the entire training set using selected features and evaluated on the test set.

### 2.7. Random Forest Models

Random forest classifier was implemented using caret R package version 6.0-94 [21] with sample weighting to decrease the effect of class imbalance. Two RF model parameters were tested: (i) Mtry—number of features sampled at each tree split (from 5 to 30); (ii) Ntree—number of trees in a forest (100, 500, 1000, 2000). MRCV procedure was applied, as follows: (i) data were split into training and validation (80%/20%) subsets; (ii) RF model was fit on training data; (iii) estimation of variable importance was made (for each tree, the prediction accuracy on the out-of-bag portion of the data is recorded; then, the same is conducted after permuting each predictor variable; the difference between the two accuracies is then averaged over all trees and normalized via the standard error); and (iv) an estimation of classification threshold is made by maximizing balanced accuracy. The MRCV procedure was repeated 100 times. Next, feature ranking was generated, as follows: (i) for each feature, the average variable importance score was calculated; (ii) the elbow technique was used to extract the most relevant features for the final RF model. Finally, the model was built on the entire training set using selected features and evaluated on the test set.

### 2.8. Machine Learning Result Integration

Several approaches were applied to integrate results from both platforms. The first one was based on statistical integration proposed by Stouffer [22]. It was used for both classification probabilities of test sets as well as classification thresholds. Additionally, several common methods were tested including: (i) mean value; (ii) maximum value; and (iii) product of two probabilities. All methods are described in detail elsewhere [23]. Similarly, both classification probabilities and classification thresholds were integrated using the same method.

## 3. Results

To construct classification models, we used two different cohorts, PPPBWWRP and MOLTEST-BIS, and two types of measured features, radiomic features from LDCT images and concentrations of serum metabolites (metabolomic features). We cleaned the datasets by removing missing values and the normalization of the data (see Section 2). For radiomic features, significant differences were not observed between cohorts after data normalization (Figure 1). Better separation between benign and malignant cases was noted using radiomic features compared to metabolomic features, but in general, these two classes are not separated. The PPPBWWRP radiomic dataset revealed three clusters of data points, of which the largest one included both benign and malignant cases (Figure 1). A refined investigation revealed that one of the smaller clusters included mostly calcified nodules that resulted from historical infections or physical damages (Appendix A), while the smallest one consists of all types of benign nodules and might represent unknown technical artifacts.

### 3.1. Univariate Analysis of Metabolomic and Radiomic Studies

Many features measured in both modalities were highly skewed, showing non-normal distribution (Appendix A, Appendix A), so we used non-parametric methods for univariate analysis to compare benign and malignant cases within each modality. We ran the analysis using all available samples in each modality. After applying multiple testing corrections, we found 94 statistically significant (FDR < 0.05) features in the radiomic data, from which 44 were downregulated and 50 upregulated. However, no statistically significant features were identified in the metabolomic dataset. The mean effect size measured for top differentially regulated features was also higher in radiomic data than in metabolomic data (0.6 vs. 0.26; examples of differentially regulated features are presented in Figure 2). Many features showed similar patterns of level difference, which is due to the high correlation between them within modalities (Appendix A).

### 3.2. Development of Machine Learning Models

Before modeling, we removed highly correlated features in each modality. The MRCV procedure used for parameter tuning and feature selection (Appendix A) resulted in the following models: (i) metabolomics—11 features for LR and 14 for RF; (ii) radiomics—11 features for LR and 8 for RF. Only three features were common between both ML models in metabolomics mode (PC(41:5), PC-O(42:6), and PC(42:7)) and three in radiomics mode (glcm InverseVariance, shape Flatness, and glcm Id). As can be observed in Appendix A, as well as Appendix A, the LR approach has good classification performance in training and test sets for radiomic data. However, in the case of metabolomics, RF has better performance in the test set. Next, the results from metabolomic and radiomic modalities were integrated within each ML approach using four different methods (see Section 2). Statistical and product integrations show the best results. In both, more than half of the patients in the test set were properly classified by all models after integration (Figure 3). In most cases, integration decreased the number of false positive and negative findings, giving superior results compared to a model based on only one modality.

Looking at the model performance indices on the test set after applying the estimated classification thresholds, we observed diverse results. The RF model was better than the LR model for metabolomic data; however, it was worse for radiomics mode and after statistical and product integration (Table 3). The overall performance was moderate with F1 scores ranging from 0.55 to 0.8 and AUCs ranging from 55.5% to 84.8%. ROC curve analysis showed that there is a potential to tune threshold values to meet other specific goals of a model (Figure 4). The ROC curves for the other tested integration methods are presented in Appendix A. Finally, the results of the training set showed that models were not significantly overtrained (Appendix A, Appendix A). Integrating classification results from two modalities slightly increased model performance in comparison to the basic radiomics model (e.g., F1 and BAcc), regardless of the integration method used (Table 3; Figure 4). However, this increase was not statistically significant.

## 4. Discussion

We used two different modalities to build the prediction model for early lung screening support: (i) radiomic characteristics from LDCT scans collected in a large cohort, and (ii) serum metabolome profiles collected in a study with a smaller sample size. In the univariate analysis, a pool of significant features was found for radiomics. In contrast, for metabolomics, none of the investigated serum metabolome features were significant (FDR < 0.05). This indicates that LDCT has a better ability to distinguish lung cancer in the presented study on a single feature level. Yet, the size of the cohort used for radiomics was much bigger than for metabolomics, which can impact the observed result. Out of the significant radiomic features, run percentage (RP) can be primarily distinguished, which measures the coarseness of the texture by taking the ratio of the number of runs and number of voxels in the ROI. A higher value indicates a finer texture, and in Figure 2B (first graph), we can observe much lower values for malignant cases. The second highlighted feature is long run emphasis (LRE), which measures the distribution of long run lengths, and greater value indicates longer run lengths and more coarse structural textures. For malignant cases, we observed a higher value of LRE, as expected (Figure 2B, second graph). Next, we constructed two separate models for both modalities. Again, the model built using radiomic features showed better performance compared to the metabolomics-based model. The AUC for the best radiomics model in the test set was 83% (LR model), while for metabolomics, it was only 60.3% (RF model). The better-performing metabolomics model (RF) included 14 features (namely, Histidine, Spermidine, PC(41:5), PC(42:7), PC(42:2), PC(33:4), PC-O(42:6), LPC-O(16:1), AC(0:0), AC(8:1), CE(17:0), TG(51:1), TG(51:4), and TG(44:4)). Noteworthily, a reduced level of histidine was previously noted in the sera of patients with non-small-cell lung cancer [24]. Additionally, increased concentrations of spermidine were noted in serum/plasma and urine of patients with different types of malignancies including lung cancer [25]. However, for other (lipid) components, data on specific associations with lung cancer was not determined. When radiomic features were considered, better performance was observed for the LR model with the following features: glcm InverseVariance, shape Flatness, glszm ZonePercentage, ngtdm Strength, firstorder InterquartileRange, glrlm RunLengthNonUniformityNormalized, glcm MCC, glrlm LongRunLowGrayLevelEmphasis, glcm Id, glszm LargeAreaLowGrayLevelEmphasis, and gldm LargeDependenceHighGrayLevelEmphasis. Several features were previously reported in other studies, e.g., LargeAreaLowGrayLevelEmphasis shows effective discrimination of lung cancer from tuberculosis with AUC 92% [26]. Other examples are ngtdm Strength and glszm ZonePercentage, which showed the effectiveness of the response to immunotherapy for non-small-cell lung cancer [27]. Finally, we integrated predictions from both modalities. The product integration on LR shows slightly better performance compared to a single radiomics model, with an AUC of 84.75%. Moreover, the observed NPV was the highest (80%). Yet, the highest PPV was observed for radiomic features and the LR model. 

To summarize, our results indicate that combining the outcomes of the machine learning models based on two different modalities—LDCT radiomics and serum metabolomics—slightly increases the potential of available methods to build diagnostic tools that could discriminate benign and malignant nodules detected in participants of lung cancer screening programs. However, the preliminary results of our pilot study must be extended and validated using larger cohorts from other screening studies. Hopefully, authors of future screening programs will perform metabolic and radiomic analyses and share their data. Moreover, a combination of radiomics with other molecular or genomic signatures may result in more promising outcomes. For example, the combination of miRNA classifiers with radiomic features resulted in the increased performance of pancreatic cancer diagnosis [28]. Also, a combination of radiomics with clinical characteristics may further enhance the performance of classification models, which was previously reported for the discrimination between pneumonia-like lung cancer from pulmonary inflammatory lesions [29].

## Figures and Tables

**Figure 1 biomolecules-14-00044-f001:**
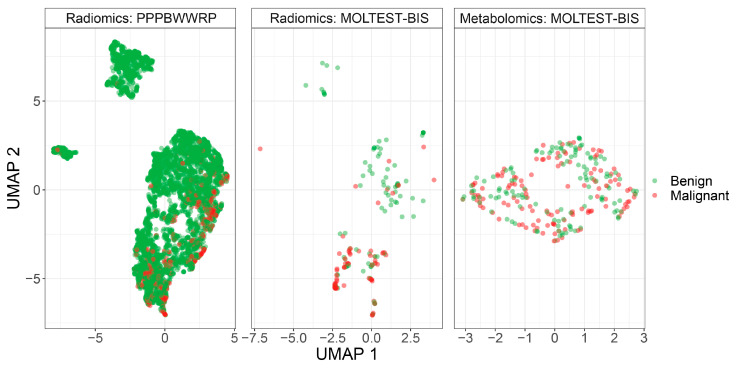
UMAP visualization of patients’ clustering using either radiomic or metabolomic modalities.

**Figure 2 biomolecules-14-00044-f002:**
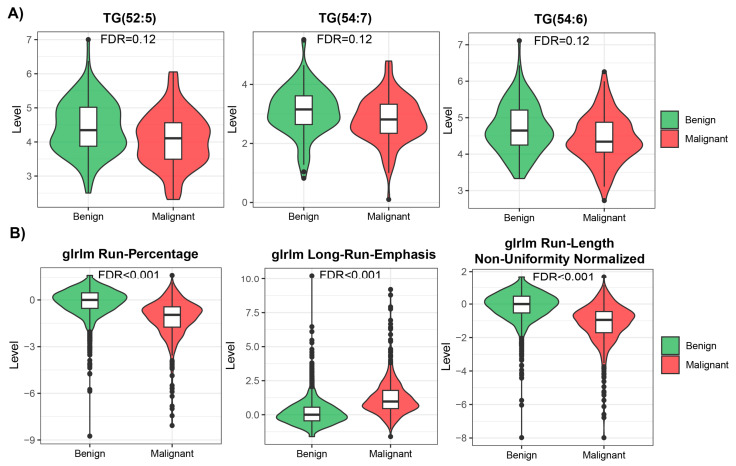
Top 3 most differentially changed features in metabolomic (**A**) and radiomic (**B**) modalities.

**Figure 3 biomolecules-14-00044-f003:**
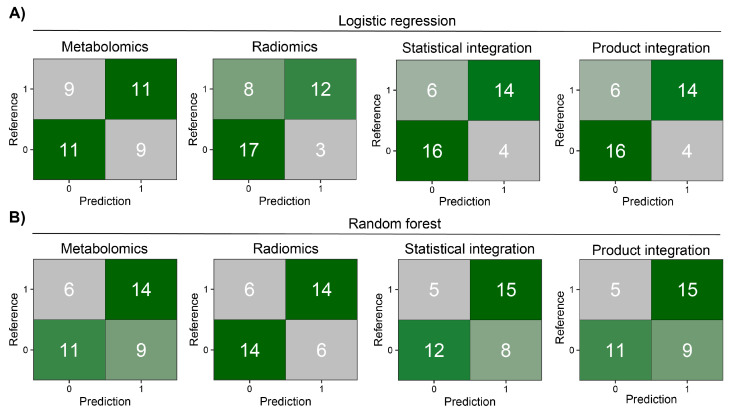
Heatmaps of confusion matrices for prediction models on the MOLTEST-BIS test set for each modality and after integration. (**A**) Results of logistic regression models. (**B**) Results of random forest models.

**Figure 4 biomolecules-14-00044-f004:**
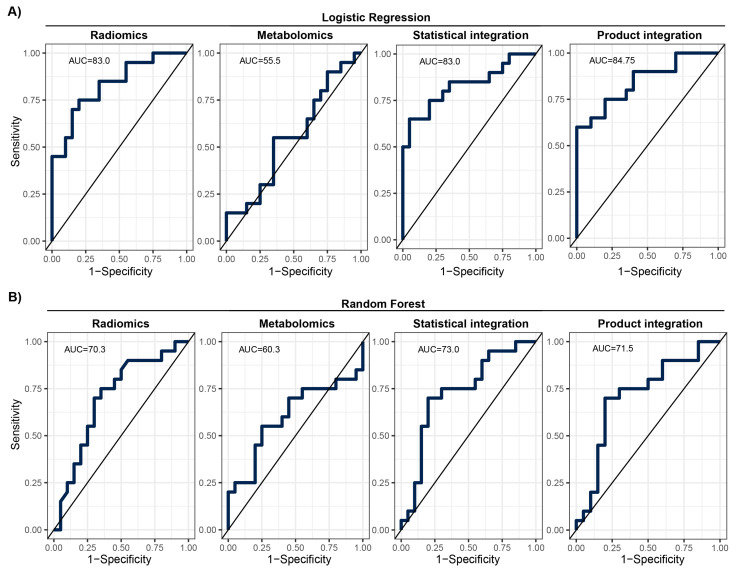
ROC with given AUC for prediction models on the MOLTEST-BIS test set for each modality and after integration. (**A**) Results of logistic regression models. (**B**) Results of random forest models.

**Table 1 biomolecules-14-00044-t001:** Basic characteristics of the analyzed population. NA means that the information was not available. Since a patient could have benign and malignant nodules at the same time, for radiomics data, n represents the number of patients with at least one nodule of a given type.

	Radiomics	Metabolomics
Benign nodules		
n	994 (75%)	123 (50%)
Screening program		
PPPBWWRP	906 (91%)	0 (0%)
MOLTEST BIS	88 (9%)	123 (100%)
Sex: male/female	445/549 (45%/55%)	66/57 (54%/46%)
Age years: (median)	NA	51–79 (67)
Smoking pack-year: range (median)	NA	26–133 (43)
Malignant nodules (lung cancer)		
n	331 (25%)	123 (50%)
Screening program		
PPPBWWRP	258 (78%)	0 (0%)
MOLTEST BIS	73 (22%)	123 (100%)
Sex: male/female	136/195 (41%/59%)	67/56 (54%/46%)
Age years: range (median)	NA	53–79 (67)
Smoking pack-year: range (median)	NA	24–138 (48)
Clinical stage:		
IA	NA	49
IB	NA	10
IIA	NA	9
IIB	NA	10
IIIA	NA	17
IIIB	NA	7
IVA	NA	16
IVB	NA	5

**Table 2 biomolecules-14-00044-t002:** Number of samples in training and test sets for ML model building.

	Radiomics	Metabolomics	Common
Train			
Benign	4569	103	68
Malignant	440	103	53
N	5009	206	121
Test			
Benign	122	20	20
Malignant	49	20	20
N	171	40	40

**Table 3 biomolecules-14-00044-t003:** Results for ML models on MOLTEST-BIS test set. The bold value shows the superiority of a particular metric and the set of data between ML approaches. LR—logistic regression, RF—random forest, CI—95% confidence interval.

Metric	Radiomics		Metabolomics		Statistical Integration		Product Integration	
	LR	RF	LR	RF	LR	RF	LR	RF
Sensitivity	0.60	**0.70**	0.55	**0.70**	0.70	**0.75**	**0.78**	0.63
Specificity	**0.85**	0.70	**0.55**	**0.55**	**0.80**	0.60	0.73	0.69
PPV	**0.80**	0.70	0.55	**0.61**	**0.78**	0.65	0.70	**0.75**
NPV	0.68	**0.70**	0.55	**0.65**	**0.73**	0.71	**0.80**	0.55
F1	0.69	**0.70**	0.55	**0.65**	**0.74**	0.70	**0.74**	0.68
Balanced accuracy	**0.73**	0.70	0.55	**0.63**	**0.75**	0.68	**0.75**	0.66
AUC (%)	**83.0**	70.3	55.5	**60.3**	**83.0**	73.0	**84.8**	71.5
AUC 95% CI	**70–96**	53–87	38–74	**42–79**	**70–96**	56–89	**73–97**	55–88

## Data Availability

The data presented in this study are available upon request from the corresponding author. The data are not publicly available due to privacy and ethical reasons.

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
