# Peer review of "Combining Low-Dose Computer-Tomography-Based Radiomics and Serum Metabolomics for Diagnosis of Malignant Nodules in Participants of Lung Cancer Screening Studies"

_biomolecules, 2023, doi:10.3390/biom14010044_

Round 1
Reviewer 1 Report
Comments and Suggestions for Authors
In this manuscript, Zyla et al. verified whether the combination of low-dose CT-based radiomics and serum metabolomics for diagnosis of malignant nodules in participants of two Lung Cancer Screening Programs performed by the Medical University of Gdansk. Generally, these findings seem to be interested. However, I have some comments.
1. Basic characteristics of the analyzed population should be presented in detail, including age, sex, ethics, race, smoking history, Lung Cancer Screening Programs, etc.
2. The 271 metabolomics features and 107 radiomics features should be presented as supplementary materials.
3. Although authors discussed the limitation of this study, the results must be validated using larger external cohort from other screening studies. A combination of radiomics with clinical characteristics should conducted to further enhance the performance of classification models.
Reviewer 2 Report
Comments and Suggestions for Authors
The manuscript, "[Combining low-dose CT-based Radiomics and serum metabolomics for diagnosis of malignant nodules in participants of Lung Cancer Screening studies" by Zyla J and Marczyk M et al, is a commendable effort to collect vast LDCT data and serum metabolome profiles collected from a study with smaller sample size, and further combined the outcomes using machine learning models.
Though only two variables, radiomics, and Metabolomics were used in this study, the authors admit their shortcomings of difference in sample sizes and inclusion of more variables that can build tools for early diagnosis or to discriminate between benign and malignant nodules in lung cancer screening.
This article is well written, describing the role of radiomics, metabolomics, and machine learning for better lung cancer diagnosis.
This article can be accepted in the current format.
Yet, I am just curious to know why this entire article and dataset are already available in the following sites:
1. https://paperswithcode.com/paper/combining-low-dose-ct-based-radiomics-
2. chromextension://efaidnbmnnnibpcajpcglclefindmkaj/https://arxiv.org/ftp/arxiv/papers/2311/2311.12810.pdf
